# Layer-polarized ferromagnetism in rhombohedral multilayer graphene

Wenqiang Zhou[1,2,5], Jing Ding[1,2,5], Jiannan Hua [1,2,5], Le Zhang[1,2], Kenji Watanabe [3], Takashi Taniguchi [4], Wei Zhu [1,2] ✉ & Shuigang Xu [1,2] ✉

Flat-band systems with strongly correlated electrons can exhibit a variety of phenomena, such as correlated insulating and topological states, unconventional superconductivity, and ferromagnetism. Rhombohedral multilayer graphene has recently emerged as a promising platform for investigating exotic quantum states due to its hosting of topologically protected surface flat bands at low energy, which have a layer-dependent energy dispersion. However, the complex relationship between the surface flat bands and the highly dispersive high-energy bands makes it difficult to study correlated surface states. In this study, we introduce moiré superlattices as a method to isolate the surface flat bands of rhombohedral multilayer graphene. The observed pronounced screening effects in the moiré potential-modulated rhombohedral multilayer graphene indicate that the two surface states are electronically decoupled. The flat bands that are isolated promote correlated surface states in areas that are distant from the charge neutrality points. Notably, we observe tunable layer-polarized ferromagnetism, which is evidenced by a hysteretic anomalous Hall effect. This is achieved by polarizing the surface states with finite displacement fields.

In rhombohedral multilayer graphene, the low-energy electrons are entirely concentrated on the two surface layers, while the bulk states exhibit an energy gap[1–6]. This distinctive characteristic provides an ideal platform for the exploration of diverse surface states. The surface electronic bands of rhombohedral graphene can be approximately described by $E \sim \pm p^N$ in a two-band model, where $E$ is the kineic energy, $p$ the momentum, and $N$ the layer number[6,7]. With increasing $N$, these surface bands become flat at low energy. Due to the instability of electronic interactions endowed by their large density of states (DOS), these surface flat bands hypothetically host strongly correlated states, such as spontaneous quantum Hall states[8], high-temperature superconductivity[1], ferromagnetism[9–12]. Furthermore, in rhombohedral multilayer graphene, the low-energy surface states, characterized by alternating intralayer and interlayer hopping, is a tailor-made

simulator of one-dimensional topological insulator in the Su–Schrieffer–Heeger model[13]. Moreover, the chiral stacking in rhombohedral graphene gives rise to large momentum-space Berry curvatures, accompanied by a giant intrinsic magnetic moment inherited from the multivalley features of graphene. This feature positions it as a promising platform for exploring topological non-trivial states, such as anomalous Hall effect (AHE)[2]. Experimentally, with recent advancement of technique for producing hexagonal boron nitride (h-BN) encapsulated structures[14], correlation-driven insulating states, magnetic states, and superconductivity have been reported in bilayer ($N = 2$)[15], rhombohedral trilayer ($N = 3$)[4,5,16–18], tetralayer[19], pentalayer[20,21], and multilayers ($N \geq 7$) graphene[3,22].

The power-law energy dispersion in rhombohedral multilayer graphene suggests that the low-energy surface flat bands are

[1]Key Laboratory for Quantum Materials of Zhejiang Province, Department of Physics, School of Science, Westlake University, 18 Shilongshan Road, Hangzhou 310024 Zhejiang Province, China. [2]Institute of Natural Sciences, Westlake Institute for Advanced Study, 18 Shilongshan Road, Hangzhou 310024 Zhejiang Province, China. [3]Research Center for Electronic and Optical Materials, National Institute for Materials Science, 1-1 Namiki, Tsukuba 305-0044, Japan. [4]Research Center for Materials Nanoarchitectonics, National Institute for Materials Science, 1-1 Namiki, Tsukuba 305-0044, Japan. [5]These authors contributed equally: Wenqiang Zhou, Jing Ding, Jiannan Hua. ✉e-mail: zhuwei@westlake.edu.cn; xushuigang@westlake.edu.cn

connected to highly dispersive high-energy bands. Consequently, the observations of strong correlations in intrinsic rhombohedral graphene have been restricted to very low carrier density ($n$) regimes ($n \rightarrow 0$)[3–5]. To isolate these surface flat bands from the high-energy dispersive bands is not only beneficial for exploring correlated states in high $n$ regimes, but also indispensable for recurring Chern band. One versatile approach for achieving this isolation is through the stacking of van der Waals materials with a twist angle and/or a lattice mismatch, which constructs moiré superlattices at two-dimensional (2D) interfaces[23,24]. These moiré superlattices impose a long-range periodic potential, resulting in band folding and the formation of a mini-Brillouin zone. This process typically leads to bandwidth reduction, thereby enhancing the effects of electron correlations. Consequently, many unique band structures emerge at low energy near the Fermi surface, accompanied by the appearance of exotic states, such as spontaneous symmetry breaking[25], superconductivity[26–28], correlated insulating states[29], orbital magnetism[30,31], Hofstadter butterfly[32,33], and topological Chern insulators[34–36].

Here, we introduce moiré superlattices into rhombohedral multilayer graphene, to separate the low-energy surface flat bands away from high-energy dispersive bands. These moiré superlattices were constructed by crystallographically aligning rhombohedral multilayer graphene with h-BN during the van der Waal assembly. Thanks to the small lattice mismatch ($\delta \approx 1.6\%$) between graphene and h-BN, a moiré superlattice can be formed with a long-range wavelength given by $\lambda = \frac{(1+\delta)a_G}{\sqrt{2(1+\delta)(1-\cos\theta)+\delta^2}}$, where $a_G = 0.246$ nm is the in-plane lattice constant of graphite, and $\theta$ the relative misalignment angle between the two lattices. Our band calculations confirm the presence of an isolated surface flat band at the conduction band, as shown in Fig. 1g and Supplementary Fig. 15. The application of large interlayer potential gives rise to remarkably layer-polarized states, indicated by our calculated results of DOS distribution in Fig. 1b. To probe the electronic transport of rhombohedral graphene, we have employed a dual-gate structure, as depicted schematically in Fig. 1d, which enables us to independently control $n$ and displacement field ($D$).

## Results

### Phase diagram and correlated states

Our devices were fabricated through the mechanical exfoliation of natural graphite. We chose rhombohedral heptalayer (7 L) graphene as the building block since our previous work indicates that it preserves the bulk characteristics of graphite meanwhile exhibiting strong correlations[3]. Similar results can be found in rhombohedral multilayer graphene with other layer numbers (see Supplementary Information). Raman spectra and mapping techniques were employed to identify the stacking order and select rhombohedral (also described as ABC) domains for device fabrication (see Fig. 1c and Supplementary Fig. 2). Figure 1h shows low-temperature ($T = 50$ mK) longitudinal ($R_{xx}$) and Hall ($R_{xy}$) resistances as a function of $n$, with carriers concentrated at one of the surfaces under a fixed $D = 1$ V nm$^{-1}$. Besides the peak at charge-neutrality point ($n = 0$), $R_{xx}$ exhibits two additional prominent peaks at high-density region. The corresponding $R_{xy}$ exhibits sign reversals, indicative of Fermi-surface reconstruction[37]. This phenomenon can be attributed to either band folding caused by the moiré superlattice or strong correlations, which we will discuss in detail later. In either case, with the assistance of the moiré superlattice, we have succeeded in isolating the surface band from high-energy band, resulting in the opening of a band gap in high $n$ regions.

To reveal the electronic transport behavior influenced by the moiré potential in rhombohedral 7 L graphene, we also fabricated a device using intrinsic rhombohedral 7 L graphene without alignment with h-BN (device D1) for reference. Figure 2a, b show color maps of $R_{xx}(n,D)$ for devices without and with moiré superlattice, respectively. In the absence of moiré, two distinct insulating states emerge at

$n = 0, D = 0$, and $n = 0, |D| > 0.4$ V nm$^{-1}$ as illustrated in Fig. 2a. This behavior closely resembles what has been observed in rhombohedral nonalayer (9 L) graphene[3]. The insulating state at $|D| > 0.4$ V nm$^{-1}$ is attributed to the opening of an energy gap in the surface states, resulting from inversion symmetry breaking induced by a large electric field. Differently, the insulating state at $n = 0, D = 0$ cannot be explained in a single-particle picture and is believed to be a correlated gap as a result of spontaneous symmetry breaking favored by surface flat band[8]. It's noted that the insulating states at $n = 0, D = 0$ strongly rely on the electronic coupling between top and bottom surfaces, which only occurs in thin-layer (roughly $N \leq 10$) rhombohedral graphene[3]. In rhombohedral multilayer graphene, this correlated gap is highly reproducible and has been observed in multiple devices (see Supplementary Fig. 7 and Supplementary Fig. 8).

Introducing moiré superlattice into rhombohedral 7 L graphene significantly modifies its transport behavior, as shown in Fig. 2b (device D2). First, the correlated gap at $n = 0, D = 0$ disappears, indicating moiré potential at the interface between h-BN and graphene effectively decouples the two surface states. Second, the critical field ($D_c$), above which a band insulator gap is opened, increases to approximately 0.8 V nm$^{-1}$. Applying $D$ via asymmetric dual gates generates a potential difference between the two surfaces, resulting in a carrier redistribution that strongly screens out the external field. The larger $D_c$ in Fig. 2b indicates the moiré potential favors carrier localization at the surfaces, thus enhancing the screening effect. This enhanced screening effect is further evident from the presence of a series of horizontal and vertical lines in the region below $D_c$, when plotting $R_{xx}$ as a function of $n_t$ and $n_b$ (see Supplementary Fig. 4). It serves to electronically decouple the two surface states and suppress their interactions, which explains the absence of correlated state at $n = 0, D = 0$. These features collectively indicate that moiré potentials at the surfaces favor the decoupling of the two surface layers in rhombohedral 7 L graphene and the emergence of layer-polarized states.

Third, we also observed additional gap states at large $D$ beyond charge-neutrality point ($n \neq 0$). When $|D| > |D_c|$, the finite band overlap between conduction and valence bands is lifted due to inversion symmetry breaking. The surface states become fully polarized, such that charge carriers concentrate on only one of the two surfaces. Namely, for positive $D > D_c$, only electrons (holes) in the conduction (valence) band at the top (bottom) layer contribute to the conductance. The screening effect vanishes, manifested as both gates are effective, accompanied by the disappearance of the horizontal and vertical lines in Supplementary Fig. 4a. Unlike device D1 in Fig. 2a, device D2 exhibits additional resistance peaks at $n_1 = 1.0 \times 10^{12}$ cm$^{-2}$ and $n_2 = 2.1 \times 10^{12}$ cm$^{-2}$ for $D > 0$. Similar extra prominent peaks also appear for $D < 0$, but at slightly different $n$. Notably, when comparing these features to those in device D1 without moiré, the peaks appearing at non-zero $n$ stem from the formation of moiré minibands. The observation of remarkable Brown–Zak oscillations[29,33,38], as shown in Fig. 2c, further confirms the formation of moiré superlattices in Device D2. In Fig. 2c, there are two distinct sets of oscillatory behavior periodic in $1/B$, which indicates that our device has doubly aligned configuration[39,40]. From the oscillation period, we can extract two twist angles $\theta_1 = 0.88°$ and $\theta_2 = 0.90°$ at the two interfaces. With this, we can assign $n_1$ and $n_2$ to the quarter filling ($\nu = 1$) and half filling ($\nu = 2$) of the moiré miniband, respectively.

The double alignment is consistent with that two sets of extra peaks appear at both $D > 0$ and $D < 0$. The possibility of double alignment can be further confirmed by the optical image of the stack, which shows the alignment between the straight edges of h-BN and graphene flakes (see Supplementary Fig. 3). Since in this system the two surface states are electronically decoupled, which is supported by the aforementioned several features, it's reasonable to treat the two moiré superlattices as independence and disregard the super-moiré effects. In the following part, we mainly focus on the $n > 0$, $D > 0$ region, namely

on the conduction band modulated by the moiré superlattice at the top layer. Similar behavior can be found in the other regions (see Supplementary Fig. 11).

The temperature dependence of resistance peaks at $\nu = 1$ and $\nu = 2$ exhibits typical insulating behavior, where the resistance increases as the temperature decreases. These insulating states at partial fillings are correlated insulators, arising from strong electron-electron interactions induced by spontaneous symmetry breaking[25], facilitated by the further flattening of surface bands through zone folding. From the Arrhenius plots in Fig. 2e and Supplementary Fig. 9, we can estimate the size of single-particle gap ($\nu = 0$) and correlated gaps at $\nu = 1$ and $\nu = 2$. The sizes of the gaps at v = 0, 1, and 2 are approximately $12.91 \pm 0.35$ meV, $4.74 \pm 0.28$ meV, and $0.81 \pm 0.03$ meV, respectively.

## Screened Landau quantization

One remarkable feature of rhombohedral 7 L graphene, compared with thinner one, lies in its decoupled surface states, which can be further evident from Landau quantization at high $B$. Figure 3 shows the Landau diagrams at $B = 14$ T of rhombohedral 7 L graphene both without and with moiré superlattices. In both devices, a series of horizontal and vertical features emerge at small $D$, which arise from the coexistence of two surface states with a high carrier density, effectively screening out their influence on each other. As $D$ increases above a critical value, the surface states become polarized, with carriers concentrating on only one of the two surfaces. In this regime, the carriers can be effectively tuned by both gates, resulting in the appearance of diagonal Landau levels (LLs) features at high $D$. The prominent

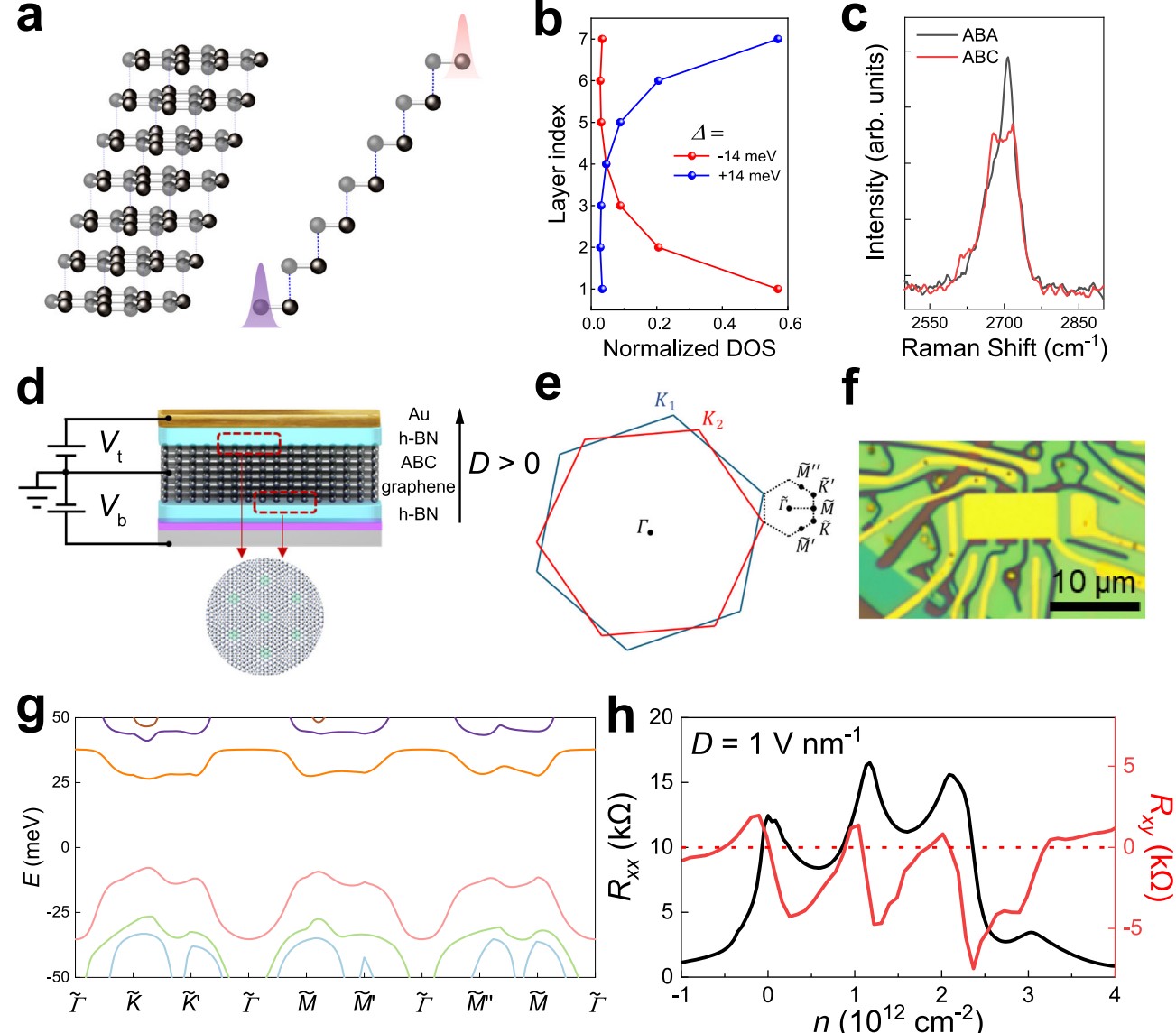

**Fig. 1 | Rhombohedral 7 L graphene moiré superlattice. a** Schematic of rhombohedral 7 L graphene. Left and right represent side view and cross-section view along the in-plane armchair direction, respectively. The two curves in the right schematic illustrate that the wavefunctions of low-energy states concentrate at the sublattices located at each surface. **b** Calculated conduction-band DOS distribution in each layer at two fixed interlayer potentials (14 meV and −14 meV). The DOS is normalized to 1. The positive interlayer potential is defined as the bottom layer having lower energy than the top layer, corresponding to $D > 0$. **c** Raman spectra of ABA-stacked and ABC-stacked 7 L graphene. **d** Schematic of a dual-gate h-BN encapsulated device with moiré superlattices at the interfaces between h-BN and graphene illustrated by red rectangular regions and the circular pattern. **e** Schematic of the mini-Brillouin zone formed by graphene/h-BN moiré superlattice. The high symmetry points in the mini-Brillouin zone are labeled. **f** Optical image of a typical device with a Hall bar geometry. **g** Calculated band structure of rhombohedral 7 L graphene with moiré superlattice at both top and bottom surface. The interlayer potential used in the calculation is 12 meV. **h** Longitudinal ($R_{xx}$) and Hall ($R_{xy}$) resistances as a function of total carrier density measured at $D = 1$ V nm$^{-1}$ and $T = 50$ mK.

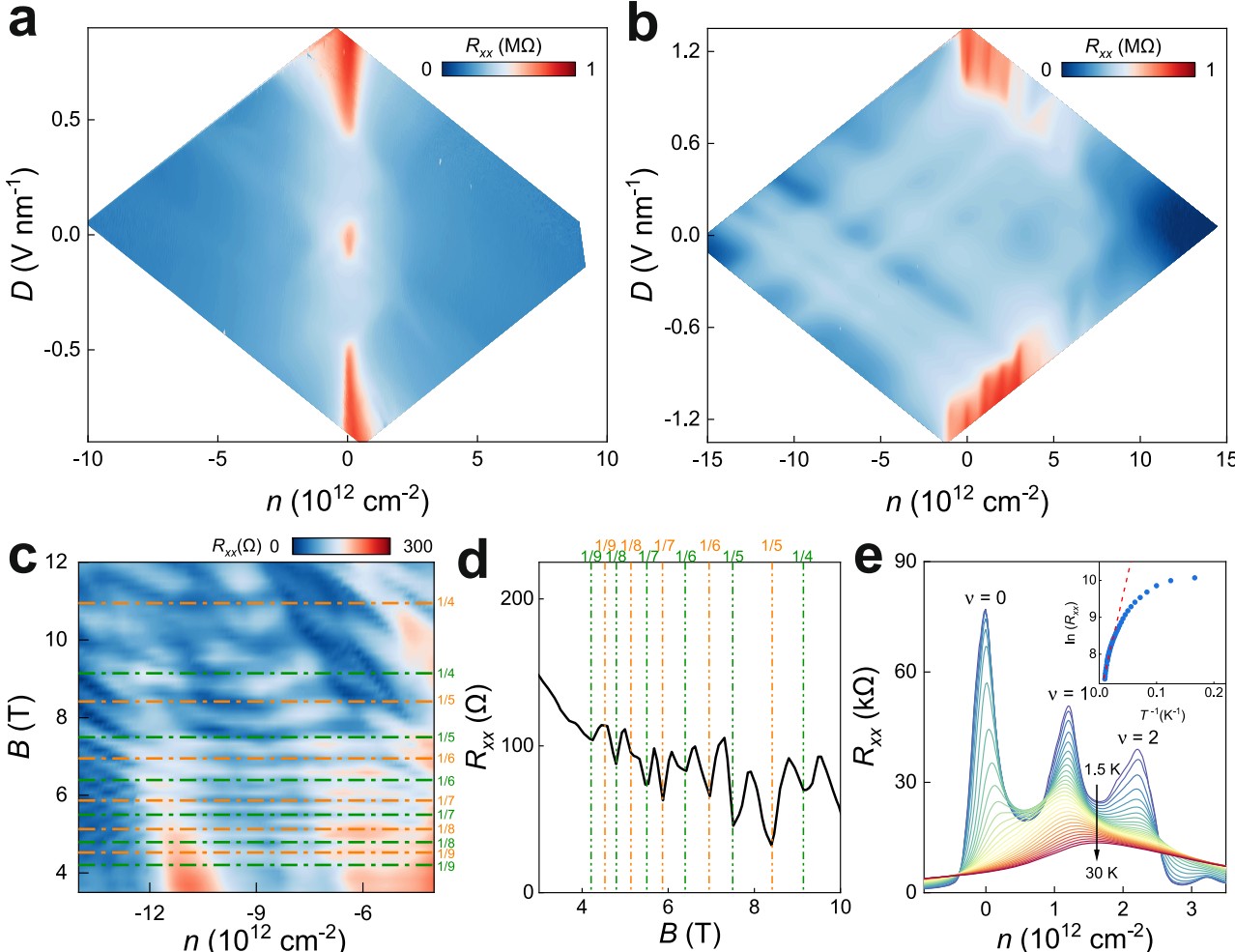

**Fig. 2 | Low-temperature transport characteristics of rhombohedral 7 L graphene without and with moiré superlattice. a, b** Color maps of longitudinal resistance $R_{xx}$ plotted in logarithmic scales as a function of carrier density $n$ and displacement field $D$ measured at $T = 50$ mK and $B = 0$ T for the devices without (**a**, device D1) and with (**b**, device D2) moiré superlattice. **c** $R_{xx}$ as a function of magnetic field $B$ and total carrier density $n$ at $T = 50$ mK and $D = 0$ V nm$^{-1}$. Quantum oscillations independent of $n$ were observed. The labels on the y-axis denote the $\frac{\phi}{\phi_0} = 1/q$, when the integer number $q$ of superlattice unit cells is commensurate with the magnetic flux quantum $\phi_0$. **d** Resistance as a function of magnetic field by

cutting the line at $n = -9.9 \times 10^{12}$ cm$^{-2}$ in **c**. The dashed lines denote the selected rational values of the magnetic flux filling of the moiré unit cell. **e** Temperature dependence of $R_{xx}$ as a function of $n$ at $D = 1.1$ V nm$^{-1}$, $B = 0$ T. Inset: Arrhenius plot (ln$R_{xx}$ versus $T^{-1}$) for charge-neutrality point ($\nu = 0$) at high-temperature region. The dashed line represents the linear fit, from which the transport gaps $\Delta$ can be extracted by ln$R_{xx} \propto \Delta/2k_BT$. The linear fits give $\Delta = 12.91 \pm 0.35$ meV, $4.74 \pm 0.28$ meV, and $0.81 \pm 0.03$ meV at $\nu = 0$, 1, and 2, respectively. The standard errors are generated during the linear fits. The data in (**c-e**) were acquired in the sample with moiré superlattice (device D2).

screening features observed in Landau diagrams resemble those seen in Bernal (ABA)-stacked graphite, albeit with opposite distributions[41–43] (see detailed discussions in Supplementary Information).

## Layer-polarized ferromagnetic states

Further insights into the broken symmetry can be gained from the Hall resistance $R_{xy}$. Figure 4a, b provide high-resolution maps of $R_{xx}$ and $R_{xy}$ in the vicinity of the correlated insulating states at $D > 0$. Near $\nu = 1$ and $\nu = 2$, maxima in $R_{xx}$ are accompanied with rapid sign changes in $R_{xy}$, indicating a change of carrier type. These sign reversals result from Fermi-surface restructuring driven by correlations and the formation of a new band edge, similar to that in twisted graphene[37]. Additionally, the evolution of $R_{xy}$ as a function of $n$ and $D$ at low $B$ in Fig. 4b exhibits gradual sign changes within the $n$ filling from $\nu = 0$ to $\nu = 1$ and $\nu = 1$ to $\nu = 2$. These sign changes correspond to divergences in $n$ and are associated with saddle points in the energy dispersion at the Fermi surface, known as van Hove singularities (vHSs)[44].

When the Fermi energy approaches vHSs, the large DOS may lead to Fermi-surface instabilities, potentially giving rise to various exotic

phases, such as superconductivity, ferromagnetism, and charge density waves. One particular example is the ferromagnetic instability, governed by the Stoner criterion[45]: $UD_F > 1$, where $U$ is the Coulomb energy, $D_F$ the DOS at the Fermi energy. The highly tunable vHSs, as shown in Fig. 4b, allow us to observe the Stoner ferromagnetism. Figure 4c, d display the $\nu$ and $D$-dependent anti-symmetrized Hall resistance $\rho_{xy}$ (see Methods) when sweeping the out-of-plane $B$ back and forth between −25 mT and 25 mT. At $\nu = 2.08$ and $D = 0.96$ V nm$^{-1}$, $\rho_{xy}$ exhibits normal linear behavior and remains independent of the sweep direction. But within a large region, $\rho_{xy}$ displays a remarkable AHE accompanied by hysteresis loops. The hysteresis becomes narrower with increasing $B$ and vanishes above a coercive field of $B_c = 7$ mT. At $B = 0$, $\rho_{xy}$ shows a non-zero value with the sign depending on the sweep direction of $B$, indicating the presence of remanent magnetization in the sample. These series of features are the hallmark of ferromagnetism, stemming from spontaneous time-reversal symmetry breaking within this system. We note that the observed hysteresis here is different from that in our previous work on intrinsic rhombohedral graphite near $n = 0$ and $D = 0$, where the hysteresis originates from

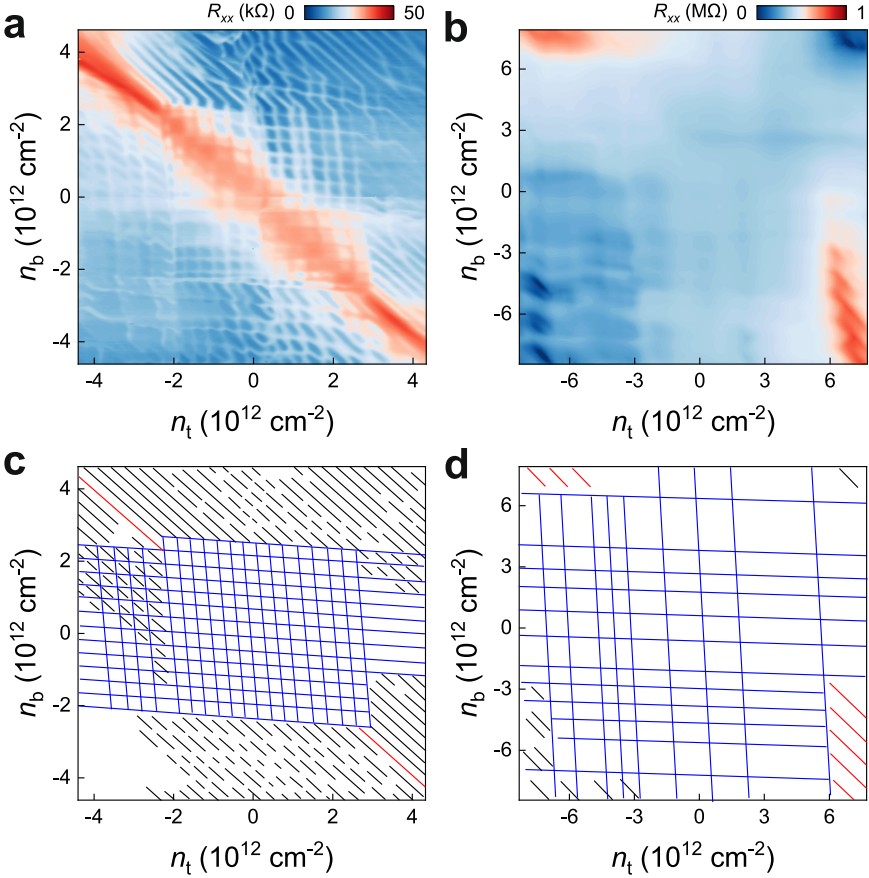

**Fig. 3 | Landau quantization in rhombohedral 7 L graphene. a, b** Color maps of longitudinal resistance $R_{xx}$ as a function of top and bottom gate-induced carrier density $n_t$ and $n_b$, measured at $B = 14$ T. The data were taken from the devices without (**a**, device D3 at $T = 50$ mK) and with (**b**, device D2 at $T = 1.5$ K) moiré superlattice. **c, d** Landau diagram depicting the LLs according to the raw data in **a** and **b**, respectively. The red lines represent the insulating states at zero field. The black lines represent LLs from polarized surface states, manifested as diagonal lines tunable by both gates. The blue lines represent screened LLs, where two surface states are strongly screened by each other, manifested as a series of horizontal and vertical features.

electronic phase inhomogeneities[3]. Whereas, in the present system, strong interactions and a large DOS within the low-energy surface flat band are responsible for the emergence of ferromagnetism. Furthermore, the hysteresis displays no Barkhausen jumps upon sweeping $B$, a phenomenon often seen in twisted graphene systems[30,31], indicating the cleanness of the graphene/h-BN moiré superlattice system.

The Hall signal comprises both a linear component originating from the normal Hall effect and an anomalous component arising from the magnetization. After subtracting the linear component, we plot the anomalous residual resistance $\Delta\rho_{xy}^{AH}$ as a function of $\nu$ and $D$, shown in Fig. 4e, which reflects the evolution of the remanent magnetization strength. The $\Delta\rho_{xy}^{AH}$ values, whether positive or negative, are marked by red and blue colors, respectively, with the intensity of the colors representing the magnitude of AHE. From Fig. 4e, obviously the AHE is highly tunable by $n$ and $D$, with the largest values appearing in the vicinity of vHSs.

Figure 4f shows the temperature-dependent hysteresis loops for a p-type-like carrier. The hysteresis of $\rho_{xy}$ disappears above a critical temperature, which further confirms the phase transition from ferromagnetism to paramagnetism. The Curie temperature, defined by the onset of hysteresis, is 4 K at an optimized position.

## Discussion

The ferromagnetism observed in rhombohedral multilayer graphene moiré superlattice differs from those previously reported in other graphene system[4,15,17,30]. First, our system exhibits a pronounced layer-polarized surface states as aforementioned. Ferromagnetism in our

system occurs only when electrons are entirely localized at one of the surface layers by applying high $D$. Namely, ferromagnetism observed here arises from electron interactions within individual surface layer. We refer to this as layer-polarized ferromagnetism. Second, Stoner ferromagnetism other than Chern band governs the AHE observed in our system. On the one hand, the emergence of ferromagnetism instabilities in our system spans a wide range, including non-integer moiré band fillings, and is enhanced near vHSs within the flat moiré bands. In contrast, in twisted bilayer graphene and twisted monolayer-bilayer graphene, the observed ferromagnetism typically occurs in a narrow region near an insulator at odd-integer filling[30,46]. On the other hand, the residual $\Delta\rho_{xy}^{AH}$ near $B = 0$ in our system is relatively small (a few hundred Ohm), far from the quantized value of $h/e^2$. Our band calculations show the first moiré conduction band in our system has zero Chern number (see Supplementary Fig. 17). Such trivial band can be further confirmed by applying the Streda's formula $\frac{\partial n}{\partial B} = C\frac{e}{h}$ to experimentally observed Landau fan diagrams (see Supplementary Fig. 13). Third, the ferromagnetism in our system is exclusive to the conduction band, which is consistent with the calculated band structure in Fig. 1g showing a narrow-isolated conduction band. This contrasts with the ferromagnetism observed in the valence band of rhombohedral trilayer moiré superlattice[17].

Our results promote the observation of ferromagnetism in thick-layer graphene systems. The emergence of layer-polarized ferromagnetism is facilitated by the presence of flat surface bands, favored by both band structures of intrinsic rhombohedral graphene and the moiré superlattice. This work establishes rhombohedral multilayer

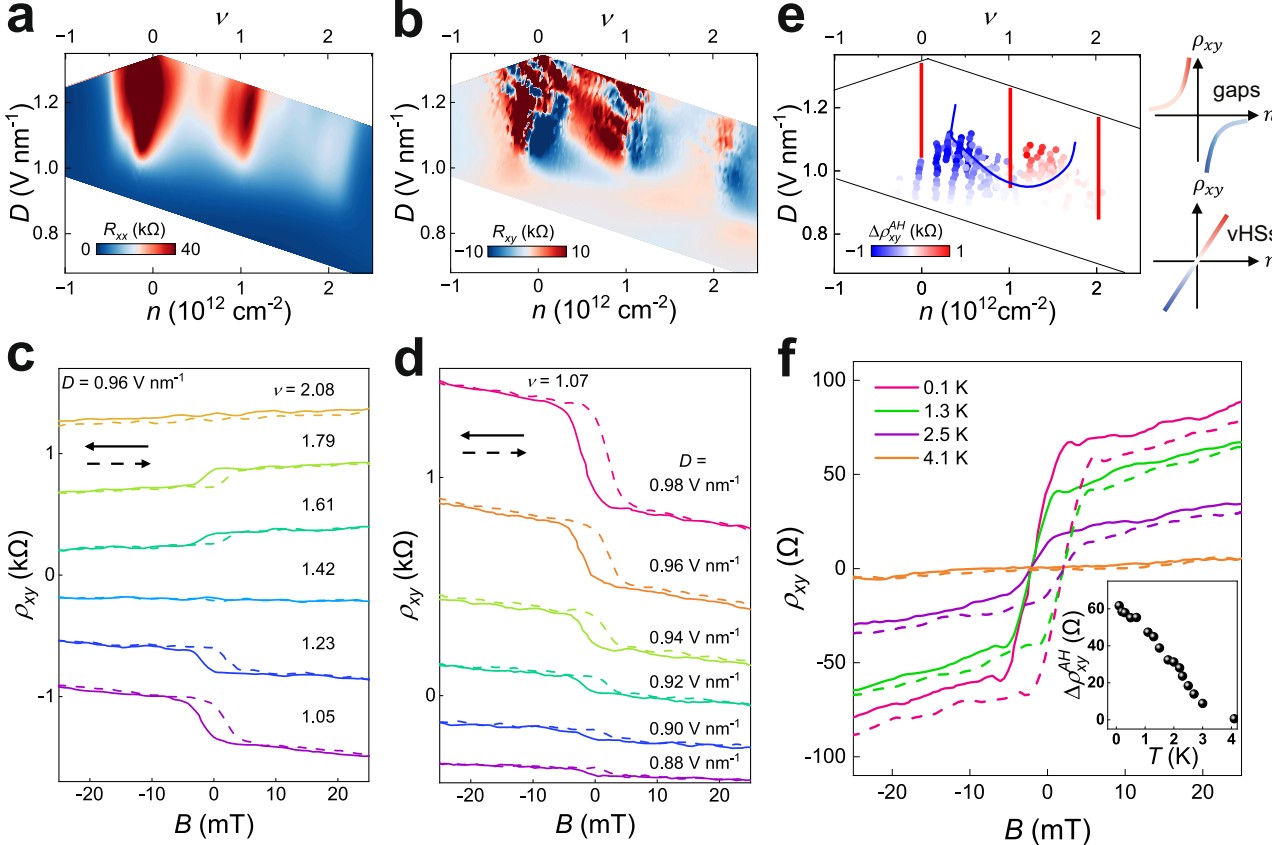

**Fig. 4 | Tunable layer-polarized ferromagnetism in rhombohedral 7 L graphene moiré superlattice. a** Fine maps of $R_{xx}$ plotted on a linear scale as a function of $n$ and $D$ near the conduction band modulated by the moiré superlattice at the top surface for $B = 0$. **b** Corresponding anti-symmetrized Hall resistance $R_{xy} = \frac{R_{xy}(+B)-R_{xy}(-B)}{2}$ at a fixed small magnetic field $B = \pm 1$ T. The Hall resistance changes its sign at gap states and vHSs in different ways. **c**, **d** Anti-symmetrized Hall resistance $\rho_{xy}$, defined in the Methods, as a function of $B$ swept back (solid arrows) and forth (dashed arrows) at **c** a fixed $D = 0.96$ V nm$^{-1}$, varying $\nu$ from 1.05 to 2.08, and **d** a fixed $\nu = 1.07$, varying $D$ from 0.88 V nm$^{-1}$ to 0.98 V nm$^{-1}$. The absolute

values are manually offset for clarity. **e** Color plots of residual resistance $\Delta\rho_{xy}^{AH}$ as a function of $n$ and $D$. The individual dots were extracted from the measurements of AHE at corresponding $n$ and $D$. The colors represent the values of $\Delta\rho_{xy}^{AH}$. The red and blue curves denote two types of sign reversal of $R_{xy}$ near gaps and vHSs, respectively, which are sketched in the right schematic. **f** Temperature dependence of AHE. $\rho_{xy}$ as a function of $B$ swept back and forth, showing ferromagnetic hysteresis at different temperatures, with fixed $\nu = 1.97$ and $D = 0.91$ V nm$^{-1}$. The inset shows the evolution of residual resistance $\Delta\rho_{xy}^{AH}$ as a function of temperature. All the data of **a**–**f** were taken in device D2 at $T = 50$ mK.

graphene as a fertile platform for exploring exotic surface states. The surface flat band in rhombohedral multilayer graphene moiré systems, when interplaying with non-trivial topological electronic states, may give rise to exotic correlated and topological physics, such as surface superconductivity[1] and quantum anomalous Hall effect in 3D systems. The Chern number in rhombohedral multilayer graphene moiré superlattice can be effectively tuned by twist, displacement field, and layer number. The fabrication technology of rhombohedral graphene can be simply applied to other layers. The tunability of the layer number in rhombohedral graphene provides great potential for observing amazing quantum states. For example, during the preparation of this manuscript, the observation of fractional quantum anomalous Hall effect in rhombohedral pentalayer graphene has been reported[47].

## Methods
### Device fabrication
We fabricated high-quality rhombohedral 7 L graphene using h-BN encapsulated structures with the assistance of the dry transfer method. The stacking order domains within the multilayer graphene were identified by Raman spectroscopy (WITec alpha300). During the dry transfer process, ABC stacking domains often shrink or even entirely convert to ABA stacking. To enhance the success rate, we isolated ABC domains from ABA domains by cutting the flake with a tungsten tip manipulated under a microscope. We

found the cutting process did not significantly alter the domain distribution (see Supplementary Fig. 2). The entirely isolated ABC flake can survive after being encapsulated by h-BN (see Supplementary Fig. 3b). Subsequently, we picked top h-BN and ABC graphene in sequence using a PDMS-supported PC film. The h-BN/graphene heterostructure was then released onto a bottom h-BN exfoliated on a 285 nm SiO$_2$/Si substrate in advance, forming the final stack.

To fabricate moiré superlattice devices, we intentionally align the crystallographic axes of graphene and h-BN by utilizing their straight edges. Typically, exfoliated large flakes of both graphene and h-BN, being hexagonal lattices, exhibit straight edges along their easy cleavage plane (either zig-zag or armchair). Supplementary Fig. 3a shows an optical image of the final stack for device D2. Notably, one of the natural cleavage edges of graphene is oriented perpendicular to the two straight edges of both the top and bottom h-BN, indicating that this stack is likely to be doubly aligned. Given the indistinguishable zig-zag and armchair edges, the aligned angle between graphene and h-BN can be around either 0° or 30°, which can be easily distinguished from transport data.

To verify the alive ABC domains in the final stack, we further characterized it by Raman spectroscopy as shown in Supplementary Fig. 3b. Pure ABC domains were carefully selected for the device design, with particular attention to regions devoid of bubbles, as determined through atomic force microscopy.

For the electrical contacts, we patterned the electrodes by e-beam lithography and selectively etched the top h-BN using $CHF_3/O_2$ plasma by controlling the etch duration. With this procedure, the multilayer graphene was exposed for metal deposition, thus forming 2D surface contacts. The electrodes and metallic top gates were fabricated by standard e-beam lithography and e-beam evaporation. The device was finally shaped into Hall bar geometry through drying etching with $CHF_3/O_2$ plasma.

**Electronic transport measurements**

Low-temperature transport measurements were performed in a dilution fridge (Oxford Triton) with a base temperature down to 50 mK. To minimize electronic temperature effects, all the wires were filtered by RC and RF filters (available from QDevil) in the mixing chamber. Standard low-frequency AC measurement was used to simultaneously obtain longitudinal and Hall resistances of the Hall bar device through lock-in amplifiers (SR830) operating at a frequency of 17.77 Hz. To measure the fragile ferromagnetic states, the AC current was limited to 5 nA. For other measurements, the current was increased to 100 nA to enhance signal quality. Gate voltages were applied using Keithley 2450 or 2614B.

The dual-gate structure of our devices provides independent control over both the total carrier density $n = n_b + n_t = \frac{C_b \Delta V_b}{e} + \frac{C_t \Delta V_t}{e}$ and the displacement field $D = \frac{C_b \Delta V_b - C_t \Delta V_t}{2\varepsilon_0}$, where $\Delta V_b = V_b - V_b^0$ ($\Delta V_t = V_t - V_t^0$) is the effective bottom (top) gate voltage, $V_b$ ($V_t$) the applied bottom (top) gate, $n_b$ ($n_t$) the bottom (top) gate-induced carrier density, $V_b^0$ ($V_t^0$) the offset voltage, $C_b$ ($C_t$) the bottom (top) gate normalized capacitance measured from Hall effect at normal states, $e$ the elementary charge, and $\varepsilon_0$ the vacuum permittivity.

The measured Hall resistance $R_{xy}$ inevitably contains signals from longitudinal resistance $R_{xx}$ due to the unperfect geometry. To remove the components of $R_{xx}$ from $R_{xy}$, we used the standard procedure to anti-symmetrize Hall resistance ($\rho_{xy}$) by $\rho_{xy}(B, \leftarrow) = [R_{xy}(B, \leftarrow) - R_{xy}(-B, \rightarrow)]/2$ and $\rho_{xy}(B, \rightarrow) = [R_{xy}(B, \rightarrow) - R_{xy}(-B, \leftarrow)]/2$, where $\leftarrow$ and $\rightarrow$ represent the swept magnetic field from positive to negative and from negative to positive, respectively. The residual resistance is defined as $\Delta \rho_{xy}^{AH} = [\rho_{xy}(B = 0, \leftarrow) - \rho_{xy}(B = 0, \rightarrow)]/2$.

## Data availability

Relevant data supporting the findings of this study are available within the article and the Supplementary Information file. All raw data are available from the corresponding authors upon request.

## Code availability

The codes supporting the findings of this study are available from the corresponding authors upon request.

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

## Acknowledgements

This work was funded by National Natural Science Foundation of China (Grant No. 12274354, S.X.), the Zhejiang Provincial Natural Science Foundation of China (Grant No. LR24A040003, S.X.; XHD23A2001, S.X.), the R&D Program of Zhejiang province (2022SDXHDX0005, W.Z.), and Westlake Education Foundation at Westlake University. We thank Chao Zhang from the Instrumentation and Service Center for Physical Sciences (ISCPS) at Westlake University for technical support in data acquisition. We also thank the Westlake Center for Micro/Nano Fabrication and the Instrumentation and Service Centers for Molecular Science for facility support. K.W. and T.T. acknowledge support from the JSPS KAKENHI (Grant Numbers 21H05233 and 23H02052) and World Premier International Research Center Initiative (WPI), MEXT, Japan.

## Author contributions

S.X. conceived the idea and supervised the project. W. Zhou fabricated the devices. J.D. built up the measurement system. W. Zhou performed the transport measurement with the assistance of J.D. and L.Z. J.H. and W. Zhu performed the band structure calculations. K.W. and T.T. grew h-BN crystals. S.X. wrote the paper with input from W. Zhou, W. Zhu, and J.H. All authors contributed to the discussions.

## Competing interests

The authors declare no competing interests.
