## [Peer Review File · Nature Communications]

Layer-polarized ferromagnetism in rhombohedral multilayer grapheneReviewers' Comments:

Reviewer #1:

Remarks to the Author:

In this manuscript, the authors performed an electronic transport study of rhombohedral hepta-layer graphene Moiré superlattices. By applying electrical displacement field, the electrons can be polarized close to one surface of the sample, where they observed correlated insulating states and ferromagnetic states, similar to those observed in angle-twisted graphene systems.

Both correlated electronic phases and Moiré superlattice-modulated surface states were previously reported in multilayer graphite. In Ref [3] of the manuscript, correlated phases in rhombohedral graphite were first observed, although this work mostly focused on the charge neutrality point. In Ref [43], The surface and bulk states of hexagonal multilayer graphite Moiré superlattice was studied. The new observation made in this manuscript is that when a large displacement field is applied to polarize the carriers to near one surface of the rhombohedral graphite, correlated insulating states like those in twisted graphene systems as well as ferromagnetic phases can be induced when the graphite is aligned to the hBN lattice. I think this gives this manuscript novelty to be considered publishing in Nature Communication, although it would be much better if the results can be generalized to rhombohedral graphite with different layer numbers, instead of focusing on 7L-graphene -- a very specific system. In addition to that, I have the following questions and comments that I would like the authors to address.

1. The 'surface state' referenced in this manuscript corresponds to the situation where electrons are mostly polarized to one side of the system by an external displacement field. But normally a surface state (or edge state, in two dimensional systems) corresponds to the electronic states that spontaneously localized at the surfaces (or edges) of the system, such as those in quantum Hall systems and in three-dimensional topological insulators. I am not sure if the observed ferromagnetism observed in this manuscript can be called "surface ferromagnetism" since it relies on the large displacement field. The other phrase "layer-polarized ferromagnetism" used in the manuscript, although less pronounced, seems more appropriate in this situation.
2. The authors observed magnetic hysteresis at a few points in the density-displacement phase space and showed they disappear at high temperature. I agree the data set is solid evidence of ferromagnetism. However, the relation between the emergence of ferromagnetism and the emergence of correlated gaps is not clear. Does the ferromagnetism disappear at the D-fields where the correlated gaps are not observed and where the layer polarization is not significant? The manuscript can benefit from additional magnetic hysteresis data so that the boundary of the ferromagnetic phase can be (at least roughly) mapped out in the phase diagram shown in Fig. 2(b).
3. The hysteresis loops shown in Supplementary Fig. 10 and Supplementary Fig. S11 did not show closed loops. It seems the forward and backward scans simply has an overall shift. Could the authors rule out the possibility the hysteretic behavior is from the instrument, (trapped flux, for example) instead of the sample?
4. In Fig.2(c), the authors showed magnetic oscillations that are not sensitive to carrier density and attribute them to the two sets of Brown-Zak oscillations corresponding to the Moiré superlattice formed with the two hBN flakes. Firstly, these results seem less pronounced compared to the oscillation patterns reported in Ref. 3. Some patterns are not clear and have large deviations from the prediction (the dashed lines). And there is a significant portion of the oscillation patterns that are sensitive to the carrier density. In Ref. 3, the data were obtained at relative high temperatures to prevent the interference of Landau quantization. Does the same problem also apply here since these data were measured at quite large magnetic field? Secondly, the data were collected at zero displacement field so that both sets of Brown-Zak oscillations should appear. Is it possible to measure at large and opposite D fields so that the two sets of oscillations can be individually resolved? That

would be a stronger support of the existence of two sets of Moiré superlattice. Finally, in Supplementary Fig. 6a, some oscillation patterns are also nearly independent of density, which I believe, are from the SdH oscillations of complex Fermi surfaces. Could the authors rule out the possibility that data in Fig. 2(c) have a similar origin?

5. The authors determined whether the carriers were fully polarized by detecting the opening of a bulk gap at the charge neutrality point. Could the authors estimate the density distribution in each layer when such phase is induced? I think calculating the real space wavefunction distribution of the flat bands as a function of interlayer potential will be helpful.

6. In Fig. 1e, I would like the authors to add a diagram showing the mini-Brillouin zone and the high symmetry points that were used to label the x-axis of Fig. 1e.

Reviewer #2:

Remarks to the Author:

The work by W. Zhou et al reports the observation of a ferromagnetic state in rhombohedral 7-layer graphite doubly aligned to boron nitride. The authors attribute the observed phenomena to Stoner ferromagnetism. Ferromagnetism has been observed in rhombohedral graphite (RG) previously (e.g. Ref 3 in main text), but the reported phenomena here is distinct in that it is associated with a moire reconstructed flat band that is localized to one surface. I believe that the results are interesting and are worthy of publication in Nature Communications, pending a response to the questions I have below.

My main point of contention is that the authors stress that the ferromagnetic state is different than what has been observed in RG and twisted bilayer graphene before, and is likely associated with a ferromagnetic instability governed by the Stoner criterion. However, the phenomena reported here reminds me of what has been observed in twisted mono-bilayer graphene (.e.g M. He et al Nature Comm 12, 4727 (2021)). In twisted mono-bilayer graphene it is observed that there is a non-quantized anomalous Hall effect (AHE) away from integer fillings of the moire unit cell in the symmetry broken metallic phases surrounding correlated insulating states. The AHE is typically attributed to orbital magnetism associated with spontaneous valley polarization. Can I understand the observed phenomena reported here in the same way, or is there some crucial signature that distinguishes the observed phenomena in this work as Stoner ferromagnetism? To this end, I have a few questions:

Can the authors determine the Chern number of the surface band out of which the ferromagnetic state is observed? Can the authors provide Landau fans taken at constant D cutting through the insulating states at $\nu=1$ and 2 ? Does the insulating state slope in $n-B$ according to the Streda formula with a well defined Chern number? Further, are there quantum oscillations that project to $\nu=1$ and 2 at $B=0T$?

A few smaller points:

I am not convinced by the gap size determination for the $\nu = 2$ state. In Supp. Fig. 8d, the resistance changes only very slightly over the range of measured temperatures and does not show clear thermal activation behavior (doesn't really look linear at any point). Further in R_{xy} , there is not a sharp change in the sign of the Hall density like there is at $\nu=0$ and $\nu=1$ (Fig. 4b).

Do you observe AHE at negative displacement field? Since the structure is fairly symmetric (moire interfaces at both sides) I would assume there would be nearly identical surface states for both signs of D .

The assignment of dual alignment is not particularly clear from Fig. 2c. Can the authors provide either a line cut or some kind of averaged signal that shows resistance dips or conductance peaks at distinct sequences of rational flux fillings? Something like Fig. 2a of Reference 43 or Fig. 3d of Reference 44?

Reviewer #3:

Remarks to the Author:

The authors of the manuscript "Surface ferromagnetism in rhombohedral heptalayer graphene moiré superlattice" report on transport measurements of rhombohedral heptalayer (7L) graphene aligned with hBN. The presence of the hBN alignment gives rise to a formation of moiré superlattice. This, in turn, gives rise to mini-bands with reduced bandwidth and enhanced density of states that promote the appearance of electronic instabilities. The authors carry out measurements of ρ_{xx} and ρ_{xy} using a dual-gated geometry, allowing tuning both charge density and a displacement field between the layers in situ. The claim of the manuscript is the observation in the high displacement field regime of an anomalous Hall effect with a hysteresis loop, indicating the presence of ferromagnetic behavior. The authors argue that the appearance of this ferromagnetic behavior is due to a Stoner-like instability. Experimental measurements are motivated with a tight-binding bandstructure plot, but no additional theoretical modeling of transport behavior is included. The authors also include measurements of unaligned heptalayer samples to highlight the role of moiré superlattice.

I find the manuscript interesting and overall well-written. I believe Nature Communications is a good fit for this work, but I ask the authors to answer the following questions/make changes to the manuscript, which I think can help with the overall message and readability of the manuscript:

- I find discussing the "3D" nature of the rhombohedral heptalayer very confusing. A good example is the discussion of the first paragraph on page 5. The authors highlight the role of screening and argue that the system behaves as a 3D system. What do the authors mean? Do they mean that a momentum along the z-axis is a well-defined quantum number, and one has a Thomas Fermi screening along the z-axis? As far as I understand, as the authors also highlight throughout the manuscript, the authors mean that in a high displacement field, charges that comprise the flat band of interest become primarily localized only on the top and bottom layers and decouple from one another. This is not 3D behavior to me. The authors highlight the role of layer-polarization/surfaces. As such, I would ask the authors to remove the references to 3D nature or strongly rephrase them to avoid confusion. In fact, I would refer to it as layer polarization - in fact, authors already include that phrasing in the manuscript - that quenches the kinetic energy of the carriers promoting interaction-driven instabilities. The manuscript is already interesting on its own, and there is no need, in my opinion, to have this additional language of invoking "surface states", "3D nature", etc. Given the authors' introduction, I imagine that they were trying to connect the reasoning to the surface states of 3D topological insulators.

- As the authors emphasize, the role of layer polarization is important. This is schematically shown in Fig. 1a. Given that the authors have a continuum model implemented, could the authors compute from the model layer polarization (by measuring probability density on each layer) of states in the flat band of interest as a function of displacement field? This would also help understand the extent of screening involved.

- In the case of Fig. 2b. COuld the authors also plot on top of the box on the x-axis filling per moiré unit cell? This could help identify the origins of the features as coming either from single-particle or interaction-driven gaps.

- On that note, the last sentence of the first paragraph of pg. 6, just before "Screened Landau quantization," is confusing. Could the authors write each gap's values explicitly rather than as a single sentence?

- On pg. 3, in the paragraph on "Phase diagram and correlated states" authors invoke sign reversals in R_{xy} as indicating Fermi surface reconstructions. This analysis typically holds well, provided no multiple Fermi surface pockets exist. Can the authors comment on this? In particular, the authors carried out SdH measurements (and FFT of the signal) in supplemental figures. Based on the frequencies in that analysis, can the authors conclude the number and degeneracy of the Fermi pocket, or is the SdH data not fine enough for the FFT? I would encourage authors to comment on this.

- As a theorist, the most exciting question is the interpretation of the origins of ferromagnetic behavior. The authors make a cryptic comment that "Stoner ferromagnetism other than Chern band governs the AHE observed in our system". Do the authors mean by this that the flat band they find from the continuum model in Fig. 1e has a zero Chern number? Can the authors comment on this by evaluating the Chern number from their model? If indeed it is zero, can the authors comment/speculate about the possible origins of ferromagnetism?

Minor:

- I would encourage authors to revisit the language of a few paragraphs. For example, the paragraph starting "The double alignment..." on pg. 5 reads a bit confusing with phrases such as: "As our graphene is sufficiently thicker.." etc.

- While plotting ρ_{xy} in Fig. 4e is perfectly valid, in moiré literature, people typically plot Hall density so $\nu_{\text{hall}} \sim (d R_{xy}/dB)^{-1}$ (See, for example, Fig. 2 in <https://www.nature.com/articles/s41586-021-03192-0>). Could the authors consider making a similar plot in supplemental figures?

Point-by-point responses to the reviewers' comments

We thank the reviewers for taking the time to assess our manuscript and raising constructive suggestions to improve it. We have carefully considered all the comments and revised the manuscript accordingly. We believe that this letter and the revised manuscript fully addressed all their valuable comments. Point-by-point responses to all the comments are as follows:

Reviewer #1 (Remarks to the Author):

In this manuscript, the authors performed an electronic transport study of rhombohedral hepta-layer graphene Moiré superlattices. By applying electrical displacement field, the electrons can be polarized close to one surface of the sample, where they observed correlated insulating states and ferromagnetic states, similar to those observed in angle-twisted graphene systems.

Both correlated electronic phases and Moiré superlattice-modulated surface states were previously reported in multilayer graphite. In Ref [3] of the manuscript, correlated phases in rhombohedral graphite were first observed, although this work mostly focused on the charge neutrality point. In Ref [43], The surface and bulk states of hexagonal multilayer graphite Moiré superlattice was studied. The new observation made in this manuscript is that when a large displacement field is applied to polarize the carriers to near one surface of the rhombohedral graphite, correlated insulating states like those in twisted graphene systems as well as ferromagnetic phases can be induced when the graphite is aligned to the hBN lattice. I think this gives this manuscript novelty to be considered publishing in Nature Communication, although it would be much better if the results can be generalized to rhombohedral graphite with different layer numbers, instead of focusing on 7L-graphene -- a very specific system. In addition to that, I have the following questions and comments that I would like the authors to address.

Response: We thank the reviewer for the thorough summary of the background and recognizing the novelty of our work. We agree that it would be much better to extend our system to a general rhombohedral graphite with different layer number.

According to the reviewer's suggestion, we have fabricated more devices beyond 7L graphene. Although it requires significant experimental efforts to make and measure additional devices, we succeeded to fabricate a device of rhombohedral 6L graphene and to measure it in detail (see Fig. R1). In this device, we observed similar layer-polarized insulating states at large displacement field, strong correlated gaps near zero displacement field, as well as the half-metal and quarter-metal states. Although this device is non-aligned, the new data demonstrate the universal correlated effects in intrinsic rhombohedral multilayer graphene owing to its flat surface band. We believe that the aligned rhombohedral multilayer graphene hosts similar strong correlations as that in 7L graphene. Thereby, we changed the title to "Layer-polarized ferromagnetism

in rhombohedral multilayer graphene moiré superlattice”.

In the revised manuscript, Fig. R1 has been added as Supplementary Figure 8.

Figure R1 | Phase diagram and spontaneous symmetry breaking in rhombohedral 6L graphene. **a**, Color plot of n - D mapping at $B=0$ T. **b**, $\sigma_{xx}(n, B)$ mapping at fixed $D=0$ V nm^{-1} . We observed pronounced $\nu = -12$ corresponding to the gap between zeroth LL and first LL in the valence band, consistent with the layer number of 6. **c**, $R_{xx}(n, D)$ mapping at $B=4$ T. Quantum oscillations with degeneracies of 4, 2, 1 were observed. The lift of degeneracy reveals spontaneous symmetry breaking occurs at half-metal ($\Delta\nu=2$) and quarter-metal states ($\Delta\nu=1$).

1. The ‘surface state’ referenced in this manuscript corresponds to the situation where electrons are mostly polarized to one side of the system by an external displacement field. But normally a surface state (or edge state, in two dimensional systems) corresponds to the electronic states that spontaneously localized at the surfaces (or edges) of the system, such as those in quantum Hall systems and in three-dimensional topological insulators. I am not sure if the observed ferromagnetism observed in this manuscript can be called “surface ferromagnetism” since it relies on the large displacement field. The other phrase “layer-polarized ferromagnetism” used in the manuscript, although less pronounced, seems more appropriate in this situation.

Response: We thank the reviewer’s valuable suggestions and agree that “layer-polarized ferromagnetism” is more suitable to describe our observations. Thereby, we have changed the title to “Layer-polarized ferromagnetism in rhombohedral multilayer graphene moiré superlattice”. In the abstract and main text, we also withdrew the claim of “surface ferromagnetism” and replaced it by “layer-polarized ferromagnetism”.

2. The authors observed magnetic hysteresis at a few points in the density-displacement phase space and showed they disappear at high temperature. I agree the data set is

solid evidence of ferromagnetism. However, the relation between the emergence of ferromagnetism and the emergence of correlated gaps is not clear. Does the ferromagnetism disappear at the D -fields where the correlated gaps are not observed and where the layer polarization is not significant? The manuscript can benefit from additional magnetic hysteresis data so that the boundary of the ferromagnetic phase can be (at least roughly) mapped out in the phase diagram shown in Fig. 2(b).

Response: We thank the reviewer for agreeing our solid evidence for the observations of ferromagnetism in rhombohedral multilayer graphene. Indeed, the appearance of ferromagnetism is strongly dependent on the displacement fields. We have measured a series of magnetic hysteresis curves at various positions in the n - D mapping shown in Fig. 2b. The results and additional data of magnetic hysteresis were summarized in Fig. R2. These data are included in Fig. 4e and Supplementary Figure 10 in the revised manuscript. We would like to point out that each point (296 points in total) in Fig. R2c (or Fig. 4e) was obtained by measuring individual magnetic hysteresis and calculating the residual resistance $\Delta\rho_{xy}^{AH} = [\rho_{xy}(B = 0, \leftarrow) - \rho_{xy}(B = 0, \rightarrow)]/2$. The blue or red colored points represent the pronounced anomalous hall effect with nonzero magnetic hysteresis loops, while the white points represent the absence of hysteresis loops. Obviously, pronounced ferromagnetic phase occurs only when large D is applied, and layer polarization is significant. Meanwhile, we find the ferromagnetic phase spans a wide range, including non-integer moiré band filling, and is enhanced near van Hove singularities (vHSs) within the flat moiré bands. Therefore, we believe that it is the Stone criterion that governs the ferromagnetism observed here.

Figure R2| The relation between the emergence of ferromagnetism and correlated gaps. **a**, n - D maps near regions where correlated gaps appear. **b**, the corresponding anti-symmetrized Hall resistance. **c**, the distributions of ferromagnetic states at n - D maps. **d** and **e**, additional data of magnetic hysteresis depending on the displacement field at selected n in **c**.

3. The hysteresis loops shown in Supplementary Fig. 10 and Supplementary Fig. S11 did not show closed loops. It seems the forward and backward scans simply has an overall shift. Could the authors rule out the possibility the hysteretic behavior is from the instrument, (trapped flux, for example) instead of the sample?

Response: We thank the reviewer for pointing out the possibility of the hysteretic behavior resulting from the instrument issues. Experimentally, at the regions with relatively weak ferromagnetic states, the hysteresis loops are indeed less pronounced compared with remarkable ones shown in Fig. 4. Occasionally, a slightly vertical shift may occur when sweeping magnetic field, due to the background noise. Similar experimental results can be found in the literatures, such as Fig. S9, S10 in *Nat. Commun.*, **13**, 6468, (2022) and Fig. 2c in *Nano Lett.*, **22**, 1, 238-245, (2022). Nevertheless, we can confirm the existence of anomalous hall effect in Supplementary Fig. 10 and S11 (corresponding to Supplementary Fig. 11 and Fig. 12 in the revised manuscript) from the following aspects. 1): The magnetic hysteresis loops still exist after removing the entire vertical shift induced by the background noise. Obvious hysteresis loops manifesting as horizontal shift below the coercive fields can be observed. 2): Remarkable nonlinear Hall signals can be observed when the fields are below the coercive fields. 3): The anomalous Hall effect can be observed only in specific positions in n - D mapping. In normal states (as shown in Fig. R3), we observed the normal linear Hall signals under the same instrument environment and measuring parameters.

Figure R3 | The Hall signals at normal states for **a**, Device D2 at negative D. **b**, Device D4. The linear behavior and no hysteresis loops can be observed at normal states.

In the revised manuscript, we have added the data from normal states in Fig. R3 into Supplementary Fig. 11 and 12 as the comparisons.

4. In Fig.2(c), the authors showed magnetic oscillations that are not sensitive to carrier density and attribute them to the two sets of Brown-Zak oscillations corresponding to the Moiré superlattice formed with the two hBN flakes. Firstly, these results seem less pronounced compared to the oscillation patterns reported in Ref. 3. Some patterns are not clear and have large deviations from the prediction (the dashed lines).

Response: We thank the Reviewers to point out the issues of Brown-Zak oscillations in rhombohedral graphene moiré superlattice.

Indeed, the quantum oscillations (including Brown-Zak oscillations) in rhombohedral graphite moiré superlattice are less pronounced than that in intrinsic rhombohedral graphite and Bernal-stacked graphite moiré superlattice. This is reasonable, since the existence of surface flat band, together with moiré flat band, in rhombohedral graphite moiré superlattice commonly results in a lower carrier mobility due to its large effective mass. Similar observations can be found if we compare the quantum oscillations in graphene/h-BN superlattice and magic-angle twisted bilayer graphene. For example, the Brown-Zak oscillations in magic-angle twisted bilayer graphene [see Extended Data Fig. 7 e and f in Cao et al., *Nature*, **556**, 80-84, (2018)] are much less pronounced than that in graphene/h-BN superlattice [see Fig. 2 in Kumar et al., *Science*, **357**, 6347, 181-184, (2017)].

And there is a significant portion of the oscillation patterns that are sensitive to the carrier density.

Response: We agree that the Brown-Zak oscillations emerge only in a finite interval of carrier densities in our devices. Actually, similar phenomena were also observed in Bernal-stacked graphite moiré superlattices as shown in Extended Data Fig. 3 of Ref. 43 (Mullan et al., *Nature*, **620**, 756-761, (2023)). It seems that it's a common case in various kinds of moiré superlattice, including graphene/h-BN [Kumar et al., *Science*, **357**, 6347, 181-184, (2017)], twisted bilayer graphene [Lin et al., *Nano Lett.*, **20**, 10, 7572-7579, (2020)], twisted monolayer-bilayer graphene [Xu et al., *Nat. Phys.*, **17**, 619-626, (2021)]. This is an interesting phenomenon that is worthy of further study. A very recent paper [Vries et al., *Nano Lett.*, **24**, 2, 601-606, (2024)] provides a possible explanation of this universal phenomenon. They claim that the Brown-Zak oscillations mostly appear near the Lifshitz transitions and they originate from Aharonov-Bohm interference of electron waves following a Kagome-like network in the vicinity of Lifshitz transitions in the minibands of moiré superlattice.

In Ref. 3, the data were obtained at relative high temperatures to prevent the interference of Landau quantization. Does the same problem also apply here since these data were measured at quite large magnetic field?

Response: On the one hand, Brown-Zak oscillations in moiré system is much more robust than Landau quantization (Kumar et al., *Science*, **357**, 6347, 181-184, (2017)). On the other hand, the flatband in our system gives rise to a lower carrier mobility, compared to that in Bernal-stacked graphite moiré system in Ref. 43 [Mullan et al., *Nature*, **620**, 756-761, (2023)] (We assume the reviewer probably would like us to compare our data with that in Ref. 43 instead of Ref. 3). The critical magnetic field (B_c) required to observe pronounced Landau quantization is much larger according to the simple criterion $\mu B_c > 1$, where μ is the carrier mobility. Therefore, we can observe a remarkable Brown-Zak oscillations in Fig. 2c at relatively low temperature and it can survive at quite large magnetic field, since the Landau quantization is relatively suppressed when the flat band exists.

Secondly, the data were collected at zero displacement field so that both sets of Brown-Zak oscillations should appear. Is it possible to measure at large and opposite D fields so that the two sets of oscillations can be individually resolved? That would be a stronger support of the existence of two sets of Moiré superlattice.

Response: We appreciate the reviewer’s suggestion. We indeed expected to observe single set of Brown-Zak oscillations at large D fields. However, at large D fields, the surface band becomes much more flatter. This is beneficial for the observation of correlated states, including layer-polarized ferromagnetism reported here. But, it leads to the difficulty of observing quantum oscillations due to the same reason aforementioned. Even the Brown-Zak oscillations are smeared at large displacement field, as shown in Fig. R4.

Figure R4| The Landau fan diagrams under large positive (a) and negative (b) displacement fields. No distinguishable quantum oscillations appear, which prevents us from using Brown-Zak oscillations to extract twist angles.

Finally, in Supplementary Fig. 6a, some oscillation patterns are also nearly independent of density, which I believe, are from the SdH oscillations of complex Fermi surfaces. Could the authors rule out the possibility that data in Fig. 2(c) have a similar origin?

Response: In non-aligned rhombohedral 7L graphene, the quantum oscillations indeed show unusual patterns arising from its complex Fermi surfaces (probably due to the trigonal warping effect), as shown in Supplementary Fig. 6a. However, the patterns are pretty different from those in Fig. 2c. Firstly, in the region of carrier density between $-5.5 \times 10^{12} \text{ cm}^{-2}$ and $-3.5 \times 10^{12} \text{ cm}^{-2}$, the pattern shows no horizontal lines, indicating that the quantum oscillations, though weakly, still depend on the carrier density. Secondly, there is a series of Landau level crossings, arising from the zeroth Landau level overlapping with the valence-band Landau levels. These Landau level crossings have been discussed detailedly in previous reports [Shi et al., *Nature*, **584**, 210-214, (2020); Seiler et al., *Nature*, **608**, 298-302, (2022)]. We do not observe similar features in Fig. 2c. Therefore, we believe the quantum oscillations in the aligned device (Fig. 2c) and non-aligned device (Supplementary Fig. 6a) have different origins.

In summary, we believe our device is doubly aligned sample based on the following reasons. Firstly, we observed pronounced correlated insulating states at both $D > 0$ and $D < 0$. By comparison, if the device is singly aligned, significant asymmetry in n - D maps and only very weak correlated states in nonaligned side can be observed, such as that

in singly aligned 5L rhombohedral moiré superlattice reported in Ref. 47 [arxiv:2309.17436, (2023)]. Secondly, the optical image of the van der Waals stack shown in Supplementary Fig. 3 illustrates that the straight edge of graphene is perpendicular to both the top and bottom h-BN, which indicates it's doubly aligned. Thirdly, we observed two sets of Brown-Zak oscillations in our device.

In the revised manuscript, we have added above discussion accordingly and Fig. R4 in Supplementary Figure 13.

5. The authors determined whether the carriers were fully polarized by detecting the opening of a bulk gap at the charge neutrality point. Could the authors estimate the density distribution in each layer when such phase is induced? I think calculating the real space wavefunction distribution of the flat bands as a function of interlayer potential will be helpful.

Response: We thank the reviewer's suggestions. Accordingly, we have calculated the density of states (DOS) distribution in each layer in the revised manuscript. The results are summarized in Fig. R5, in which Fig. R5a presents the DOS in each layer under different interlayer potentials and Fig. R5b presents the normalized DOS in each layer under two fixed interlayer potentials (large positive and negative values). Remarkable layer-polarized DOS distribution can be observed under high interlayer potentials.

Figure R5 | Calculated layer-dependent DOS distribution. **a**, DOS distribution in each layer as a function of interlayer potentials. **b**, Normalized DOS in each layer at large positive and negative interlayer potentials.

In the revised manuscript, Fig. R5a and R5b has been added to Supplementary Figure 16 and Figure 1b, respectively.

6. In Fig. 1e, I would like the authors to add a diagram showing the mini-Brillouin zone and the high symmetry points that were used to label the x-axis of Fig. 1e.

Response: We thank the reviewer for the helpful suggestion. Accordingly, we have added the schematic of Fig. R6 to Figure 1e in the revised manuscript.

Figure R6 | Schematic of mini-Brillouin zone of h-BN/graphene superlattice with a small twist angle. The high symmetry points in the mini-Brillouin zone are labeled.

Reviewer #2 (Remarks to the Author):

The work by W. Zhou et al reports the observation of a ferromagnetic state in rhombohedral 7-layer graphite doubly aligned to boron nitride. The authors attribute the observed phenomena to Stoner ferromagnetism. Ferromagnetism has been observed in rhombohedral graphite (RG) previously (e.g. Ref 3 in main text), but the reported phenomena here is distinct in that it is associated with a moire reconstructed flat band that is localized to one surface. I believe that the results are interesting and are worthy of publication in Nature Communications, pending a response to the questions I have below.

Response: We are grateful for the Reviewer’s positive assessment of our work and inspirational suggestions.

My main point of contention is that the authors stress that the ferromagnetic state is different than what has been observed in RG and twisted bilayer graphene before, and is likely associated with a ferromagnetic instability governed by the Stoner criterion. However, the phenomena reported here reminds me of what has been observed in twisted mono-bilayer graphene (e.g M. He et al Nature Comm 12, 4727 (2021)). In twisted mono-bilayer graphene it is observed that there is a non-quantized anomalous Hall effect (AHE) away from integer fillings of the moire unit cell in the symmetry broken metallic phases surrounding correlated insulating states. The AHE is typically attributed to orbital magnetism associated with spontaneous valley polarization. Can I understand the observed phenomena reported here in the same way, or is there some crucial signature that distinguishes the observed phenomena in this work as Stoner ferromagnetism? To this end, I have a few questions:

Response: We thank the reviewer for raising this question, which helps to clarify the mechanism of the ferromagnetic states observed in our system. The orbital magnetism observed in twisted monolayer-bilayer graphene mainly appears nearby odd-integer fillings ($\nu = 1$ or $\nu = 3$) of moiré bands, in both quantized AHE case [Polshyn et al., *Nature*, **588**, 66-70, (2020)] and non-quantized AHE case [He et al., *Nat. Commun.*, **12**, 4727, (2021); Chen et al., *Nat. Phys.*, **17**, 374-380, (2021)]. The origin of AHE in twisted monolayer-bilayer graphene is attributed to either spontaneous valley polarization [Chen et al., *Nat. Phys.*, **17**, 374-380, (2021); Polshyn et al., *Nature*, **588**,

66-70, (2020)] or time-reversal symmetry broken intervalley coherent (Q-IVC) state [He et al., *Nat. Commun.*, **12**, 4727, (2021)]. Both of the mechanisms require the topological nontrivial valley Chern bands ($C \neq 0$).

In our system, we find the ferromagnetic state is different from twisted monolayer-bilayer graphene system in terms of the following aspects. Firstly, we observed AHE occurring in a wide range including non-integer moiré band filling, not only nearby odd integer filling ($\nu = 1$) but also nearby even integer filling ($\nu = 2$). Secondly, the Chern number in our system is zero, extracted both from experimental data in Landau fan diagrams revealed by Streda formula $\frac{\partial n}{\partial B} = C \frac{e}{h} = 0$ (see Fig. R7) and our calculations.

Thirdly, we found the most pronounced AHE occurs near vHSs, which can be concluded by comparing Fig. 4b and 4e (or see Fig. R2b and R2c). Therefore, we believe it's the large density of states at the flat band, particularly near vHSs, that leads to the spontaneous spin/valley polarization. The ferromagnetism observed here is governed by Stoner criterion.

In the revised manuscript, we added the above discussion in the Discussion section and cited the above reference as Ref. 46.

Can the authors determine the Chern number of the surface band out of which the ferromagnetic state is observed? Can the authors provide Landau fans taken at constant D cutting through the insulating states at $\nu=1$ and 2 ? Does the insulating state slope in n - B according to the Streda formula with a well defined Chern number? Further, are there quantum oscillations that project to $\nu=1$ and 2 at $B=0T$?

Response: We thank the reviewer's suggestions. In the revised manuscript, we have determined the Chern number of the surface band both from experimental data and calculation results. Experimentally, as shown in Fig. R7, we have measured the Landau fan diagrams at fixed D cutting through the insulating states. We found all the insulating peaks are independent of magnetic fields in terms of position ($\frac{\partial n}{\partial B} = 0$). Therefore,

according to the Streda formula $\frac{\partial n}{\partial B} = C \frac{e}{h} = 0$, we get the Chern number $C = 0$ at all correlated states. Our theoretical calculations further confirm the zero Chern number.

Based on the zero Chern number observed at correlated states, we can attribute the origin of ferromagnetic states to the Stoner ferromagnetism, rather than the topological nature.

Figure R7 | The Landau fan diagrams at fixed $D > 0$ (a) and $D < 0$ (b). All the correlated resistance peaks show straight vertical lines, resulting to zero Chern numbers according to the Streda formula.

In the revised manuscript, we have added Fig. R7 in the Supplementary Figure 13 and the above discussion in the Discussion section.

A few smaller points:

I am not convinced by the gap size determination for the $\nu = 2$ state. In Supp. Fig. 8d, the resistance changes only very slightly over the range of measured temperatures and does not show clear thermal activation behavior (doesn't really look linear at any point). Further in R_{xy} , there is not a sharp change in the sign of the Hall density like there is at $\nu = 0$ and $\nu = 1$ (Fig. 4b).

Response: We agree that the gap feature at $\nu = 2$ is much weaker than that in $\nu = 0$ and $\nu = 1$. This is reasonable as we can find small R_{xx} (Fig. 4a) and relatively slight change in the sign of R_{xy} (Fig. 4b). Since the sizes of the gaps are strongly dependent on the displacement field [see Supplementary Fig. 9a in the revise manuscript or similar phase diagram in trilayer graphene in Chen et al., *Nat. Phys.*, **15**, 237-241, (2019)], we believe that to observe remarkable gapped states and sign change of Hall signals in $\nu = 2$, much higher displacement fields ($D > 1.2 \text{ V nm}^{-1}$) are required. This is a big challenge in the experiment due to the small dielectric constant and breakdown voltage of h-BN (1.2 V nm^{-1} is already quite large for this kind of device).

In Supplementary Fig. 9d, we try our best to use the linear region to fit the gap size. It is noteworthy that the Arrhenius fits used to extract gap are selected in temperature range above a critical temperature. At ultralow temperature, the gaps will be smeared by disorder induced broadening, governed by the mechanisms such as variable-range hopping. The extracted gap size at $\nu = 2$ is 0.8 meV, which is indeed much smaller than that at $\nu = 0$ (12.9 meV) and $\nu = 1$ (4.7 meV). Therefore, we believe the fit in Supplementary Fig. 9d is acceptable.

In the revised manuscript, we added the above discussion in the caption of Supplementary Fig. 9.

Do you observe AHE at negative displacement field? Since the structure is fairly symmetric (moire interfaces at both sides) I would assume there would be nearly identical surface states for both signs of D.

Response: Yes. We indeed observe similar AHE at negative displacement field, which is shown in Supplementary Fig. 11. Both nonlinear Hall signals and hysteresis loops can be observed at selected positions.

The assignment of dual alignment is not particularly clear from Fig. 2c. Can the authors provide either a line cut or some kind of averaged signal that shows resistance dips or conductance peaks at distinct sequences of rational flux fillings? Something like Fig. 2a of Reference 43 or Fig. 3d of Reference 44?

Response: We thank the suggestion. Accordingly, we have added the following plot (see Fig. R8) to Fig. 2d in the revised manuscript.

Figure R8 Resistance as a function of magnetic field by cutting the line at $n = -9.9 \times 10^{12} \text{cm}^{-2}$ in Fig. 2c. The dashed lines denote the selected rational values of the magnetic flux filling of the moiré unit cell ϕ/ϕ_0 .

Reviewer #3 (Remarks to the Author):

The authors of the manuscript "Surface ferromagnetism in rhombohedral heptalayer graphene moiré superlattice" report on transport measurements of rhombohedral heptalayer (7L) graphene aligned with hBN. The presence of the hBN alignment gives rise to a formation of moiré superlattice. This, in turn, gives rise to mini-bands with reduced bandwidth and enhanced density of states that promote the appearance of electronic instabilities. The authors carry out measurements of ρ_{xx} and ρ_{xy} using a dual-gated geometry, allowing tuning both charge density and a displacement field between the layers in situ. The claim of the manuscript is the observation in the high displacement field regime of an anomalous Hall effect with a hysteresis loop, indicating the presence of ferromagnetic behavior. The authors argue that the appearance of this ferromagnetic behavior is due to a Stoner-like instability. Experimental measurements are motivated with a tight-binding bandstructure plot, but no additional theoretical modeling of transport behavior is included. The authors also include measurements of

unaligned heptalayer samples to highlight the role of moiré superlattice.

I find the manuscript interesting and overall well-written. I believe Nature Communications is a good fit for this work, but I ask the authors to answer the following questions/make changes to the manuscript, which I think can help with the overall message and readability of the manuscript:

Response: We thank the reviewer's positive comments and important suggestions that help us further improve our work.

- I find discussing the "3D" nature of the rhombohedral heptalayer very confusing. A good example is the discussion of the first paragraph on page 5. The authors highlight the role of screening and argue that the system behaves as a 3D system. What do the authors mean? Do they mean that a momentum along the z-axis is a well-defined quantum number, and one has a Thomas Fermi screening along the z-axis? As far as I understand, as the authors also highlight throughout the manuscript, the authors mean that in a high displacement field, charges that comprise the flat band of interest become primarily localized only on the top and bottom layers and decouple from one another. This is not 3D behavior to me. The authors highlight the role of layer-polarization/surfaces. As such, I would ask the authors to remove the references to 3D nature or strongly rephrase them to avoid confusion. In fact, I would refer to it as layer polarization - in fact, authors already include that phrasing in the manuscript - that quenches the kinetic energy of the carriers promoting interaction-driven instabilities. The manuscript is already interesting on its own, and there is no need, in my opinion, to have this additional language of invoking "surface states", "3D nature, etc. Given the authors' introduction, I imagine that they were trying to connect the reasoning to the surface states of 3D topological insulators.

Response: We apologize for the confusion in the discussion of surface states and 3D nature in our system. In the original version, regarding "surface states", we refer to the low-energy electrons carrying wavefunctions that are localized at the top and bottom surfaces of the rhombohedral multilayer graphene. Such states are often referred as topology in the literature because they are somewhat analogous to the edge states in Su-Schrieffer-Heeger model [Kopnin et al., *Phys. Rev. B*, **83**, 220503(R), (2011); Xiao et al., *Phys. Rev. B*, **84**, 165404, (2011)], using a mapping of the 3D problem into a 1D case. And the theory predicted that such surface states in rhombohedral multilayer graphene can host many exotic phenomena, such as surface superconductivity [Koppni et al., *Phys. Rev. B*, **87**, 140503(R), (2013)] and ferromagnetism [Olsen et al., *Phys. Rev. B*, **87**, 115414, (2013); Pamuk et al., *Phys. Rev. B*, **95**, 075422, (2017)]. We think our system realizes the predicted ferromagnetism at the surface states.

Regarding "3D nature", we refer that in rhombohedral 7L graphene, the layer number is large enough such that even below the critical field (D_c), the two surface states are electronically decoupled and their interactions are suppressed, evidenced by the strong screening effects and the disappearance of the insulating states at $n=0$, $D=0$. These features are different from thin-layer rhombohedral graphene, such as trilayer graphene

[Zhou et al., *Nature*, **598**, 429-433, (2021); Chen et al., *Nat. Phys.*, **15**, 237-241, (2019)]. The 3D nature is also used to describe the surface states in thick rhombohedral graphene in the literatures [Kopnin et al., *Phys. Rev. B*, **83**, 220503(R), (2011); Xiao et al., *Phys. Rev. B*, **84**, 165404, (2011)].

Nevertheless, we fully agree the reviewer’s suggestion that “layer-polarized states” is better to describe our observations. Accordingly, in the revised manuscript, we try to avoid using the phrase of “surface ferromagnetism” and “3D nature”. Instead, we have changed the title to “Layer-polarized ferromagnetism in rhombohedral multilayer graphene moiré superlattice”. Additionally, the corresponding sentences in abstract and the main text were also revised.

- As the authors emphasize, the role of layer polarization is important. This is schematically shown in Fig. 1a. Given that the authors have a continuum model implemented, could the authors compute from the model layer polarization (by measuring probability density on each layer) of states in the flat band of interest as a function of displacement field? This would also help understand the extent of screening involved.

Response: We thank the reviewer’s suggestions. Accordingly, we have calculated the density of states (DOS) distribution in each layer in the revised manuscript. The results are summarized in Fig. R9, in which Fig. R9a presents the DOS in each layer under different interlayer potentials and Fig. R9b presents the normalized DOS in each layer under two fixed interlayer potentials (large positive and negative values). Remarkable layer-polarized DOS distribution can be observed.

Figure R9 | The layer-dependent density of states (DOS) distribution. **a**, DOS distribution in each layer as a function of interlayer potentials. **b**, Normalized DOS in each layer at large positive and negative interlayer potentials.

In the revised manuscript, Fig. R9a and R9b has been added to Supplementary Figure 16 and Figure 1b, respectively.

- In the case of Fig. 2b. COuld the authors also plot on top of the box on the x-axis filling per moiré unit cell? This could help identify the origins of the features as coming either from single-particle or interaction-driven gaps.

Response: We thank the reviewer for this valuable suggestion. Since in Fig. 2b, the data contain two sets of moiré unit cells with slightly different moiré wavelength ($D>0$ and $D<0$ correspond to the top and bottom moiré superlattice, respectively). It's not convenience to label the filling factors in Fig. 2b. Instead, we plot the corresponding filling factors on top of the x -axis in the n - D maps for $D>0$ and $D<0$ separately, as shown in Fig. R10.

It is noted that during the revision, we realize the charge-neutrality points in $D>0$ and $D<0$ sides may have slightly different offsets due to the different environment-induced carrier doping at the top and bottom surfaces. Therefore, when converting the carrier density n to the filling factor ν , we use a slightly different Δn_0 as the offset calibrations for $D>0$ and $D<0$.

Figure R10| Longitudinal resistance R_{xx} as a function of n and D for $D>0$ (a) and $D<0$ (b). The filling factors per moiré unit cell are labeled on the top of x -axis. The filling factors are calculated according to the corresponding twist angles for $D>0$ and $D<0$, separately.

In the revised manuscript, accordingly, we added the filling factors on top of the x -axis in Fig. 4 a-c and Supplementary Fig. 11a.

- *On that note, the last sentence of the first paragraph of pg. 6, just before "Screened Landau quantization," is confusing. Could the authors write each gap's values explicitly rather than as a single sentence?*

Response: We thank the reviewer's comment. We apologize for our confusing sentence. We have amended this sentence to make it clearer.

- *On pg. 3, in the paragraph on "Phase diagram and correlated states" authors invoke sign reversals in R_{xy} as indicating Fermi surface reconstructions. This analysis typically holds well, provided no multiple fermi surface pockets exist. Can the authors comment on this? In particular, the authors carried out SdH measurements (and FFT of the signal) in supplemental figures. Based on the frequencies in that analysis, can the authors conclude the number and degeneracy of the Fermi pocket, or is the SdH data not fine enough for the FFT? I would encourage authors to comment on this.*

Response: We thank the reviewer for this insight comment.

The rapid sign reversals in Hall signals are widely considered as an experiment evidence of gap opening, either due to the single particle gap or correlated gaps [He et al., *Nat. Phys.*, **17**, 26-30, (2021); Park et al., *Nature*, **590**, 249-255, (2021); Wu et al., *Nat. Mater.*, **20**, 488-494, (2021); Xu et al., *Nat. Phys.*, **17**, 619-626, (2021)]. It results from the sign change of the carrier mass on opposite sides of the band gaps. The sign reversal at $\nu = 0$ is corresponding to the layer-polarized insulating states, consistent with single-particle picture. The observed sign reversals in R_{xy} at $\nu = 1$ and $\nu = 2$, accompanied by the appearance of resistance peaks in R_{xx} , arise as a consequence of correlations restructuring the moiré bands.

The data in the Supplementary Fig. 7 were acquired from an intrinsic (non-aligned) rhombohedral 7L graphene, presenting as a comparison to that in rhombohedral moiré system. In this non-aligned system, all the states are metallic for $n \neq 0$. At high n region, due to the two spin and two valley degeneracies native to graphene systems, the normal metal states in intrinsic rhombohedral multilayer graphene have 4-fold Fermi pockets. The high-order hopping in rhombohedral graphene can induced a series of Lifshitz transitions in the Fermi sea topology at low-energy region (near $n = 0$), where complex trigonally warped Fermi pockets (such as annulus or 12-fold Fermi pockets) may exist [Zhou et al., *Nature*, **598**, 429-433, (2021)]. When applying large displacement field, the surface band in intrinsic rhombohedral graphene can be further flattened, dramatically increasing DOS near van Hove singularities. In this case, spontaneous spin or valley or both flavor polarization can occur, giving rise to half metal or quarter metal states.

In principle, the spontaneous symmetry breaking occurring in the correlated insulating states can also be revealed from the degeneracy reduction in SdH measurements or Landau fan diagrams. However, the extremely flat band in rhombohedral multilayer graphene moiré system significantly reduces the carrier mobility. As a result, experimentally it's difficult to resolve the SdH data in the Landau fan diagrams at layer-polarized states (under large displacement field) as shown in Fig. R7.

- As a theorist, the most exciting question is the interpretation of the origins of ferromagnetic behavior. The authors make a cryptic comment that "Stoner ferromagnetism other than Chern band governs the AHE observed in our system". Do the authors mean by this that the flat band they find from the continuum model in Fig.1e has a zero Chern number? Can the authors comment on this by evaluating the Chern number from their model? If indeed it is zero, can the authors comment/speculate about the possible origins of ferromagnetism?

Response: Indeed, we conclude the origins of the ferromagnetic behavior is governed by Stoner criterion based on the zero Chern number. In the revised manuscript, we have measured the Landau fan diagram across the correlated gaps. We did not observe the Chern insulator and the Chern number in the correlated gap is extracted to be zero based on Streda formula (see Fig. R7). Meanwhile, we have calculated the Chern number of the lowest conduction flat band (see Fig. R11). The result also demonstrates that Chern

number is zero.

Figure R11| Calculated Chern number of the first moiré conduction band in doubly aligned rhombohedral 7L graphene. Only interlayer potentials Δ larger than 8 meV are shown, since at low Δ the moiré flat bands are not fully isolated. The results show zero Chern number of the band at large region of twist angles and Δ . Our experimental case is in the zero Chern number region.

Since we observed ferromagnetic states at zero Chern number bands. We believe it is Stoner ferromagnetism rather than the topological nontrivial band governs the AHE in our system. This conclusion can be further confirmed by other evidence. Firstly, we observed AHE occurring a wide range, not only nearby integer fillings as that in twisted graphene system. Therefore, the ferromagnetism likely comes from itinerant electrons instead of localized electrons. Secondly, the residual $\Delta\rho_{xy}^{AH}$ near zero field in our results is relatively small (a few hundred Ohm), far away from the quantized value of h/e^2 . Thirdly, we found the most pronounced AHE occurs near van Hove singularities, which can be concluded by comparing Fig. 4b and 4e. Therefore, we believe it's the large density of states at the flat band, particularly near vHSs, that leads to the spontaneous spin/valley polarization. The ferromagnetism observed here is governed by Stoner criterion.

In the revised manuscript, we have added Fig. R7 and Fig. R11 in the Supplementary Figure 13 and Supplementary Figure 17, respectively.

Minor:

- I would encourage authors to revisit the language of a few paragraphs. For example, the paragraph starting "The double alignment..." on pg. 5 reads a bit confusing with phrases such as: "As our graphene is sufficiently thicker.." etc.

Response: We thank the reviewer for the careful reading of our manuscript and pointing out the language issues. We have corrected the issue mentioned. We have also proofread the manuscript carefully and corrected typos/grammar mistakes.

In the revised manuscript, the corresponding sentence is revised as:

“Since in this system the two surface states are electronically decoupled, which is

supported by the aforementioned several features, it's reasonable to treat the two moiré superlattices as independence and disregard the super-moiré effects”

Other language issues and typo have also been corrected, including the following changes:

1. In the origin version, Ref. 1 and Ref. 9 overlap. It has been corrected in the revised manuscript.
2. There is a typo in the definition of the sign of D in Methods section. We have corrected it and added the schematic of $D>0$ in Fig. 1d in the revised manuscript.
3. Additionally, when we make a proof reading of the manuscript, we found a mistake in Page 5 in the previous version, i.e. “Namely, for positive $D>D_c$, only electrons (holes) in the conduction (valence) band at the bottom (top) surface contribute to the conductance”. Now we have corrected it in the revised version. It should be “Namely, for positive $D>D_c$, only electrons (holes) in the conduction (valence) band at the top (bottom) layer contribute to the conductance”. Because for $D>0$, the top layer has higher electronic energy, forming the conduction band, while bottom layer has lower energy, forming the valence band. We have corrected it in the revised manuscript. Now it's consistent with our calculated results as shown in Fig. 1b. The revised analysis is also consistent with that reported in literatures, such as Gmitra et al., *Phys. Rev. Lett.* **119**, 146401 (2017); Chen et al., *Nat. Phys.* **17**, 374–380 (2021); Zheng et al., *Nature* **588**, 71–76 (2020); Zhang et al., *Nature* **613**, 268-273 (2023).

- While plotting ρ_{xy} in Fig. 4e is perfectly valid, in moiré literature, people typically plot Hall density so $\nu_{\text{Hall}} \sim (dR_{xy}/dB)^{-1}$ (See, for example, Fig. 2 in <https://www.nature.com/articles/s41586-021-03192-0>). Could the authors consider making a similar plot in supplemental figures?

Response: We thank the reviewer's kind comments. Accordingly, we replot Fig. 4b in terms of normalized Hall density $\nu_H = 4n_H/n_s$, where $n_H = -[e(dR_{xy}/dB)]^{-1}$ and n_s is the carrier density at full filling. The result is shown in Fig. R12.

Figure R12| a, Normalized Hall density ν_H as a function of n and D at $B = \pm 1$ T. **b**, Schematic of sign reversal of ν_H near gaps and vHSs.

In the revised manuscript, we add Fig. R12 to Supplementary Figure 14.

Reviewers' Comments:

Reviewer #1:

Remarks to the Author:

The authors have addressed my major concerns. I thank the authors for their careful revision of their manuscript and adding the additional data.

The data from the six-layer device is helpful to support the conclusion that the correlation phenomena generally exist in rhombohedral graphene multilayers, but since their 6-layer device is not aligned to boron nitride, I would rather not claim that their conclusion can be generalized to "multilayer graphene moiré superlattice". Otherwise, I can recommend its publication with a more focused title.

Reviewer #2:

Remarks to the Author:

The revisions to the manuscript based on the comments from the reviewers has strengthened the paper significantly in my opinion. My comments, and I believe the other reviewers' comments as well, were thoroughly addressed and additional data has been added to the main text and supplement to that effect. I think this work is of significant interest and should be accepted for publication in Nat. Comms. Thanks to the authors for their thorough response.

Reviewer #3:

Remarks to the Author:

I thank the authors for their careful revisions of the manuscript and methodological answers to all questions. I appreciate also authors' flexibility in changing the paper's title and removing all references to "surface states" as suggested by Referee 1 and myself.

I think the manuscript is ready to be published in its present form.

Point-by-point responses to the reviewers' comments

Reviewer #1 (Remarks to the Author):

The authors have addressed my major concerns. I thank the authors for their careful revision of their manuscript and adding the additional data.

The data from the six-layer device is helpful to support the conclusion that the correlation phenomena generally exist in rhombohedral graphene multilayers, but since their 6-layer device is not aligned to boron nitride, I would rather not claim that their conclusion can be generalized to "multilayer graphene moiré superlattice". Otherwise, I can recommend its publication with a more focused title.

Response: We thank the reviewer for the constructive suggestion on our paper and recommendation for the publication of our work. We have revised the title to “Layer-polarized ferromagnetism in rhombohedral multilayer graphene” according to the reviewer and editor’s suggestion.

Reviewer #2 (Remarks to the Author):

The revisions to the manuscript based on the comments from the reviewers has strengthened the paper significantly in my opinion. My comments, and I believe the other reviewers' comments as well, were thoroughly addressed and additional data has been added to the main text and supplement to that effect. I think this work is of significant interest and should be accepted for publication in Nat. Comms. Thanks to the authors for their thorough response.

Response: We thank the reviewer for recommending the publication of our work.

Reviewer #3 (Remarks to the Author):

I thank the authors for their careful revisions of the manuscript and methodological answers to all questions. I appreciate also authors' flexibility in changing the paper's title and removing all references to “surface states” as suggested by Referee 1 and myself.

I think the manuscript is ready to be published in its present form.

Response: We thank the reviewer for recommending the publication of our work.